# Construction of a recombinant vaccine expressing Nipah virus glycoprotein using the replicative and highly attenuated vaccinia virus strain LC16m8

**Shumpei Watanabe**[1,2]*, **Tomoki Yoshikawa**[2], **Yoshihiro Kaku**[3], **Takeshi Kurosu**[2], **Shuetsu Fukushi**[2], **Satoko Sugimoto**[2], **Yuki Nishisaka**[1], **Hikaru Fuji**[1], **Glenn Marsh**[4], **Ken Maeda**[3], **Hideki Ebihara**[2], **Shigeru Morikawa**[1], **Masayuki Shimojima**[2]*, **Masayuki Saijo**[2,5]

1 Department of Microbiology, Faculty of Veterinary Medicine, Okayama University of Science, Imabari, Ehime, Japan, 2 Department of Virology I, National Institute of Infectious Diseases, Musashimurayama, Tokyo, Japan, 3 Division of Veterinary Science, National Institute of Infectious Diseases, Shinjuku, Tokyo, Japan, 4 Australian Centre for Disease Preparedness, CSIRO, Geelong, VIC, Australia, 5 Public Health Office, Health and Welfare Bureau, Sapporo Municipal Government, Sapporo, Hokkaido, Japan

* s-watanabe@ous.ac.jp (SW); shimoji-@niid.go.jp (MS)

**Data Availability Statement:** All relevant data are within the manuscript.

## Abstract

Nipah virus (NiV) is a highly pathogenic zoonotic virus that causes severe encephalitis and respiratory diseases and has a high mortality rate in humans (>40%). Epidemiological studies on various fruit bat species, which are natural reservoirs of the virus, have shown that NiV is widely distributed throughout Southeast Asia. Therefore, there is an urgent need to develop effective NiV vaccines. In this study, we generated recombinant vaccinia viruses expressing the NiV glycoprotein (G) or fusion (F) protein using the LC16m8 strain, and examined their antigenicity and ability to induce immunity. Neutralizing antibodies against NiV were successfully induced in hamsters inoculated with LC16m8 expressing NiV G or F, and the antibody titers were higher than those induced by other vaccinia virus vectors previously reported to prevent lethal NiV infection. These findings indicate that the LC16m8-based vaccine format has superior features as a proliferative vaccine compared with other poxvirus-based vaccines. Moreover, the data collected over the course of antibody elevation during three rounds of vaccination in hamsters provide an important basis for the clinical use of vaccinia virus-based vaccines against NiV disease.

**Trial Registration:** NCT05398796.

## Author summary

The Nipah virus (NiV) is an emerging paramyxovirus that is widely observed in Southeast Asia and causes fatal respiratory and neurological diseases in humans and animals. The World Health Organization considers NiV diseases priority infectious disease for which an effective vaccine should be developed. LC16m8 is a highly attenuated strain of the

**Funding:** This study was supported by the Japan Agency for Medical Research and Development (AMED) (Grant No JP21fk0108080 (MSH, SW) and 22wm0325002 (SW)), Takeda Science Foundation (Grant No 2017045212 (SW) and 2021181927 (SW), Kanae Foundation for the Promotion of Medical Science (46th Asia Oceania Research program, 46-3) (SW), and JSPS KAKENHI (Grant No JP22K06016) (SW). The funders had no role in study design, data collection and analysis, decision to publish, or preparation of the manuscript.

**Competing interests:** The authors declare that they have no competing interests.

vaccinia virus, and more than 100,000 people have been vaccinated against it in Japan without severe side effects. Compared to other attenuated vaccinia virus vaccines, LC16m8 has a relatively high or modest proliferative potential in mammalian hosts, resulting in efficient induction of immune responses in humans. In the present study, we successfully generated recombinant LC16m8 expressing the surface glycoprotein of NiV and confirmed that these vaccine candidates could efficiently induce neutralizing antibodies against NiV in hamsters.

## Introduction

Nipah virus (NiV) is an enveloped virus with a non-segmented negative-stranded RNA genome that belongs to the genus *Henipavirus* in the family Paramyxoviridae [1]. NiV has two surface glycoproteins on its envelope, the G and F proteins, which are responsible for receptor binding and membrane fusion, respectively, and are considered major immunogens that induce neutralizing antibodies [2].

NiV was first discovered in 1998–99 and was identified as the causative agent of severe respiratory disease in pigs and encephalitis or respiratory disease in humans (276 recorded cases, 40% mortality) in Malaysia and Singapore [1,3,4]. Subsequent virus outbreaks have been reported annually in Bangladesh and India. In a few cases, person-to-person transmission through close direct contact has been reported [5,6]. In Southeast Asia, certain fruit bat species are natural reservoirs of the virus, leading to its wide circulation [4,7–10]. Because of the pandemic potential of the virus based on its specific features, the World Health Organization has prioritized NiV disease in the blueprint of research and development [11]. Currently, therapeutic drugs have not been approved for NiV, and vaccines are not available for humans. However, a relatively large number of vaccine candidates have been developed based on recombinant viruses using viral vectors, virus-like particles, and subunit vaccines, and their efficacies have been demonstrated in experimental animal models (summarized in a review [12]). Recently, NiV vaccines have been developed using mRNA platforms, and one is currently undergoing phase I clinical trials (ClinicalTrials.gov NCT05398796). Each candidate has advantages and disadvantages in terms of efficacy, safety, and cost-effectiveness. Among these, vaccine vectors based on poxviruses, including a vaccinia virus (VV) (NYVAC strain) and canary poxvirus, have been tested experimentally and shown to induce protection against lethal infection with NiV in hamsters [13,14].

Until the eradication of smallpox in 1980, multiple live VV strains were used as smallpox vaccines. However, these first-generation vaccines were associated with serious adverse events [15–18]. Second-generation vaccines, such as ACAM2000, were developed by growing first-generation seeds in cell culture. ACAM2000 is licensed, but available only for groups at high risk of variola virus infection. However, these drugs cause relatively frequent and severe side effects [19]. In the later stages of the eradication program, third-generation smallpox vaccines, including those based on the LC16m8 strain and modified VV-Ankara (MVA), were developed as safer candidates [20]. However, the efficacy of these vaccines in preventing smallpox in humans has not been evaluated, because eradication has been accomplished using first-generation vaccines. Both LC16m8 and MVA possess a highly attenuated phenotype and immunogenicity that confer protective efficacy against other orthopoxvirus infections [21–25]. In 2019, an MVA-based vaccine, MVA-BN (JYNNEOS), was approved by the Food and Drug Administration for the prevention of smallpox and monkeypox diseases. Following the global outbreak of monkeypox in 2022, third-generation smallpox vaccines, including MVA-BN and

LC16m8, gained attention for their potential to protect against monkeypox infections and their safety profiles as clinical vaccines [26].

In the early 1970s, the LC16m8 strain was established from the Lister strain in Japan by multiple passages in primary rabbit kidney cells at a low temperature of 30˚C, followed by a smaller pock-picking selection on the chorioallantonic membrane of an embryonated egg [23,27]. One genetic factor responsible for the attenuation of this vaccine is a defect in the expression of the B5R membrane protein [28].

Notably, LC16m8 replicates well in certain types of mammalian cells, such as primary rabbit kidney and RK13 cells [23, 27]. In contrast, MVA does not produce progeny viruses in most mammalian cells and only replicates in chicken embryo fibroblasts, possibly because MVA loses approximately 10% of the genome of the parental Ankara strain [20,27,29–31]. Compared with MVA, the growth feature of LC16m8 confers a superior growth capacity in mammalian hosts, including humans, resulting in robust humoral and cellular immune responses, while maintaining high attenuation in humans. In Japan, LC16m8 was approved as a licensed vaccine for smallpox and its safety profiles were evaluated by vaccinating more than 100,000 children, among whom serious adverse events were not reported [23,32,33]. Fourth-generation vaccine strains have been established using recombination techniques; for example, strain NYVAC was derived from the Copenhagen strain by deleting 18 non-essential genes and exhibited a similar or poorer growth profile in cell cultures compared to the MVA strain [34] and a highly attenuated phenotype. Third- and fourth-generation smallpox vaccines have also been used as promising vectors to express exogenous viral proteins [35–39]. Some of these vectors, particularly those used to establish vaccines against highly pathogenic viruses, have been evaluated as candidates for prophylactic and therapeutic vaccines at the experimental level. Recently, we demonstrated that immunization of a small animal model with recombinant LC16m8s encoding proteins of severe fever with thrombocytopenia syndrome virus (SFTSV) protected against a lethal challenge with SFTSV [39]. Among the aforementioned vector vaccine candidates, the MVA-based vector vaccine MVA-BN-Filo (Johnson & Johnson) was approved for Ebola virus disease by the European Commission in 2020 and is currently used as a booster vaccine. In this study, recombinant LC16m8 expressing the NiV surface glycoproteins (G or F glycoproteins) was constructed, and the induction of neutralizing antibodies against NiV was examined in hamsters inoculated with recombinant LC16m8s. These findings presented here will provide an important basis for the clinical use of VV-based vaccines against NiV disease.

## Materials and methods

### Ethics statement

Experiments using infectious NiV were conducted in BSL-3 and BSL-4 laboratories according to the biosafety regulations of the National Institute of Infectious Diseases (NIID), Tokyo, Japan. All animal experiments were reviewed and approved by the Animal Experiments Committee of Okayama University of Science and carried out in accordance with the Regulations for Animal Experiments of Okayama University of Science (#2019–75, 2020–57, 2021–72, 2023–1).

### Cell lines and viruses

RK13, 293T, and Vero cells were maintained in Dulbecco's modified Eagle's medium (DMEM; Wako, Osaka, Japan) supplemented with 10% fetal bovine serum (FBS; DMEM-10FBS). The LC16m8 VV strain (accession No. AY678275) [28] was propagated in RK13 cells. The infectious titer of LC16m8 in RK13 cells was determined using a standard plaque-forming

unit (PFU) assay [28]. The Malaysian NiV strain Ma-JMR-01-98 was kindly provided by Dr. Kouichi Morita (Institute for Tropical Medicine, Nagasaki University). The infectious dose of NiV was determined in Vero cells using a 50% standard tissue culture infectious dose (TCID$_{50}$) assay [40].

## Plasmids

To construct a plasmid encoding either the NiV G or F proteins of the Malaysian strain (Accession No. NC-002728) and encoding the EGFP protein driven by the early and late VV promoters, a complete gene synthesis service was used (Fasmac, Yokohama, Japan). Following gene synthesis, which included the promoter sequence and the open reading frame of G or F, each synthesized gene was digested with XhoI or double-digested with SalI and XhoI, and then ligated to the plasmid pRecB5R.1 [41], thus producing pRecB5R.1-NiV-G and pRecB5R.1-NiV-F. The plasmid pCAGGS-NiV-F, which encodes the F gene of the NiV Malaysian strain (NC-002728), was established previously [42]. The expression plasmid pCAGGS--NiV-G, which expresses the G protein of the NiV Malaysia strain (NC-002728) with amino acid substitutions (A181G/S553F/E544K), was constructed to replace the Kozak sequence (from accacc to gccacc) based on plasmid pCAGGS-G [43]. The G gene was amplified by PCR using the primer pair 5′-gagcatgcgccaccATGCCGGCAGAAAACAAGAAAGTTAGATTC-3′ and 5′-CAAGCTAGCTTATGTACATTGCTCTGGTATCTTAACCGCG-3′. The PCR product was digested with SphI and NheI, and inserted into pCAGGS [44].

## Generation of recombinant LC16m8 harboring NiV glycoproteins

Recombinant LC16m8 expressing NiV G or F protein, namely, LC16m8-G or LC16m8-F was generated from LC16m8 using a homologous recombination technique for foreign gene insertion with excisable selection markers, as previously reported [41]. Briefly, 293T cells were transfected with pRecB5R.1-NiV-G or pRecB5R.1-NiV-F using TansIT-LT1 (Mirus Bio, Madison, WI, USA), and the transfected cells were then infected with LC16m8 at a multiplicity of infection (MOI) of 0.05. The cells were cultured in DMEM-10FBS for three days. The culture medium and cells were collected and freeze-thawed thrice to prepare crude intermediate recombinant LC16m8 stocks. RK13 cells were inoculated with crude LC16m8 stock. Subsequently, the inoculum was removed, and the cells were overlaid with DMEM containing 2% FBS, 20 μg/ml mycophenolic acid (MPA, Sigma-Aldrich, St Louis, MO, USA), 250 μl/ml xanthine, 15 μg/ml hypoxanthine, and 1% agarose ME (Iwai Chemicals, Tokyo, Japan). After 2–3 days, plaques that were double-positive for mCherry and EGFP fluorescence signals were identified using a fluorescence microscope Eclipse Ts2-FL (Nikon, Tokyo, Japan). The cells constituting the double-positive plaques were collected in 100 μL of DMEM-10FBS. Plaque purification with the selected drugs (MPA, xanthine, and hypoxanthine) was repeated twice, and the plaque was stored as an intermediate recombinant virus while retaining the selection cassette (mCherry and xanthine-guanine phosphoribosyltransferase (XGPRT)). The resulting virus was further propagated in RK13 cells without the addition of the aforementioned selection drugs. In this final step, the selection cassette was self-excised from the intermediate recombinant virus by intragenomic homologous recombination between a sequence located at the end of the B5R gene and a short sequence homologous to it (between UP and U' as illustrated in Fig 1). After three freeze-thaw cycles, the supernatant including infectious recombinant LC16m8 was harvested by centrifugation at 2,000×$g$ and 4˚C for 10 min, and aliquots were stored as working stocks.

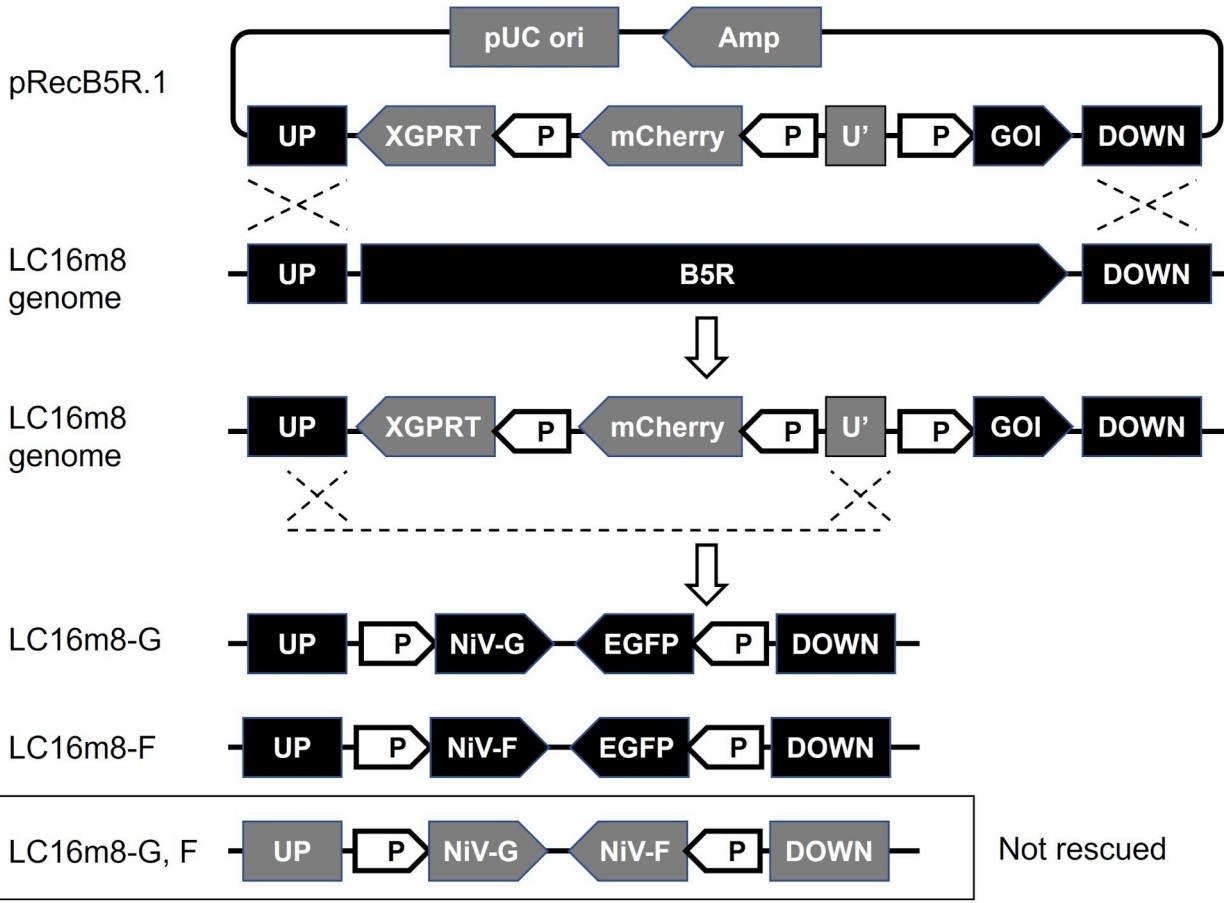

**Fig 1. Genome structure of the recombinant LC16m8 expressing NiV surface glycoproteins.** Schematic diagrams of the procedure for constructing recombinant LC16m8 (upper) and genome structures of the foreign genes inserted into the LC16m8 genome (lower). Plasmid pRecB5R.1 expresses the mCherry and XGPRT genes under the vaccinia early and late promoter (P) and contains the homologous sequences that flank the B5R gene (UP and DOWN). The genes of interest (GOIs), which were cloned into the pRecB5R.1, were transferred to the LC16m8 genome through homologous recombination. The selection marker genes XGPRT and mCherry were self-excised from the intermediate recombinant viruses by intragenomic homologous recombination between the UP and shorter sequence homologous to UP (U'). The dotted lines indicate the expected locations of the homologous recombination sites. The final recombinant viruses express the NiV G or F gene, while the complementary genome DNA expresses the EGFP gene.

### Indirect immunofluorescence assay (IFA) to confirm the expression of NiV glycoproteins

RK13 cells cultured in 6-well plates were infected with LC16m8-G or LC16m8-F at an MOI of 0.01. At 40 h post-infection (hpi), the cells were washed three times with PBS and fixed in PBS containing 2% formaldehyde. The cells were then blocked with 1% normal donkey serum (NDS) in PBS (PBS-1%NDS) overnight at 4°C. After removing the blocking buffer, cells were incubated with polyclonal rabbit serum containing anti-NiV G or anti-NiV F antibodies in PBS-1%NDS for 1 h at room temperature [42]. After three washes with PBS, the cells were incubated with Alexa Fluor 488-conjugated goat anti-rabbit IgG (H+L) (Invitrogen, Carlsbad, CA, USA) in PBS-1%NDS for 1 h at 26°C. After washing with PBS, stained cells were observed under a fluorescence microscope (Eclipse Ts2-FL).

## Animal experiments

Hamsters were used to evaluate the immunogenicity of the LC16m8-based NiV vaccine because a hamster infection model was previously established for the NiV challenge test [45]. Female 4–5-week-old Syrian golden hamsters (SLC-Japan, Shizuoka, Japan) were used at the time of the primary injection of LC16m8 or recombinant LC16m8s in all animal experiments. First, 17 hamsters were intramuscularly inoculated with 100 μl of DMEM-10FBS containing $1\times10^6$ PFU of the VV strain LC16m8 at the right hindlimb to endow preexisting immunity. Four weeks after pre-immunization with VV, three groups of hamsters (VV-G-single-site, VV-G/F-single-site, and VV-mock-single-site) (N = 4, 6, and 7) were inoculated once or twice at 2-week intervals with the viral solution containing a total of $5\times10^6$ PFU of the recombinant viruses ($5\times10^6$ PFU of LC16m8-G, a mixture of LC16m8-G and LC16m8-F ($2.5\times10^6$ PFU each), and DMED-10FBS growth medium). Second, 19 hamsters were inoculated with the LC16m8-based NiV vaccine without pre-immunization with LC16m8. Three groups of hamsters (Group G-single-site, F-single-site, and G/F-single-site) (N = 6, 6, and 7) were inoculated once or twice with the virus solution of 100 μl of DMEM-10FBS, which contained a total of $5\times10^6$ PFU of the recombinant viruses (LC16m8-G, LC16m8-F, or both viruses containing equal PFU) intramuscularly at 2-week intervals. All injections were performed under anesthesia with combination drugs (0.3 mg/kg medetomidine, 4.0 mg/kg midazolam, and 5.0 mg/kg butorphanol). Two weeks after the final dose of recombinant viruses, hamsters were euthanized using a combination of anesthetics and serum was collected.

For the evaluation of antibody induction during a series of three-time immunizations with the NiV glycoprotein-expressing LC16m8s in VV-preimmunized hamsters, at first, 27 hamsters were injected intramuscularly with 100 μl of DMEM containing $1\times10^6$ PFU of LC16m8 in the right hindlimb for pre-immunization. Four weeks after inoculation, VV-preimmunized hamsters in Group 1 (VV-G-single-site) (N = 8) and Group 2 (VV-F-single-site) (N = 9) were injected with 200 μl of DMED-10FBS containing $2\times10^6$ PFU of LC16m8-G and that of the LC16m8-F, respectively. The VV-preimmunized hamsters in Group 3 (VV-G-F-separate-sites) (N = 10) were injected with 100 μl of DMEM containing $1\times10^6$ PFU of LC16m8-G and LC16m8-F at the right and left hindlimbs, respectively. These injections were repeated 1–3 times at a 2-week interval. Two weeks after the final injection of the designated recombinant virus, serum samples were collected as described above. After each immunization with LC16m8 or recombinant LC16m8s, the clinical signs of all hamsters were monitored daily for one week.

## Measurement of antibody titers in hamster serum

Serial dilutions of hamster serum were prepared, and virus neutralization tests were performed (final volume:50 μL/well). Each sample was added at a ratio of 1:2 to the dilution media (DMEM-10FBS), and quadruplicate two-fold serial dilutions were performed. An equal volume of dilution medium containing 100 $TCID_{50}$ of NiV was added to each well and incubated for 1 h at 37°C. The virus-serum mixtures were added to confluent Vero cells seeded in a 96-well plate. The 50% neutralization dose ($ND_{50}$) for 50 μl of hamster serum was calculated according to the Behrens-Kärber method [46]. The neutralization titer of each serum sample was determined using a previously established VSV-NiV-SEAP assay [47].

To determine the IFA titers of the antibodies induced in hamster serum, BHK cells cultured in a 96-well plate were transfected with 0.1 μg of pCAGGS-NiV-G or pCAGGS-NiV-F using TansIT-LT1 (Mirus Bio). At 24–36 h posttransfection, the cells were washed with PBS three times, fixed with PBS containing 2% formaldehyde, and then blocked by PBS-1%NDS overnight at 4°C. The cells were then incubated with polyclonal rabbit serum containing anti-NiV

G or anti-NiV F antibodies in PBS-1%NDS for 1 h at room temperature [42]. After washing with PBS 3 times, the cells were incubated with Alexa Fluor 488-conjugated goat anti-hamster IgG (H+L) (Invitrogen) in PBS-1%NDS for 1 h at room temperature. After washing three times with PBS, the stained cells were observed under a fluorescence microscope (BZ-X810, Keyence, Osaka, Japan). The IFA titer was defined as the reciprocal of the highest serum dilution at which a fluorescent signal was detected.

## Statistical analysis

Data were analyzed and final graphs were prepared using GraphPad Prism software (version 9.3.1). One-tailed Student's *t*-tests (unpaired test with Welch's correction) were used to compare the means between the two groups. A P-value < 0.05 was considered significant. The number of animals in each group and specific details of the statistical tests are indicated in the figure legends.

## Results

### Generation of recombinant LC16m8s expressing NiV surface glycoproteins

We previously reported a system for constructing a recombinant LC16m8 coding foreign genes based on a homologous recombination technique in which the target sequence was inserted into the B5R region of the LC16m8 genome [41]. During the construction process, the recombinant clone harboring the gene of interest was selected by the expression of the mCherry gene under conditional drug selection with MPA, xanthine, and hypoxanthine. The cassette of selection markers (mCherry and XGPRT) was self-excised in the culture without the selection drugs to yield the final recombinant virus (Fig 1). Using this method, the complete gene coding sequence of G of the NiV Malaysia strain or NiV-F was inserted into the LC16m8 genome. Subsequently, the intermediate recombinant viruses (with the selection cassette) were independently subjected to three cycles of plaque purification using the selected drugs. After plaque purification, each of the resulting viruses was grown once in culture without the selected drugs, and each virus was stored as a working stock. In poxviruses, gene loss can occur through insertions, deletions, or point mutations [48]. To monitor the occurrence of deletion of target foreign genes or contamination with the original LC16m8 during successive cultures of the virus stock, the gene coding sequence of EGFP was inserted into the LC16m8 genomes (Fig 1). By checking the EGFP fluorescence signal, we confirmed that cytopathic effects (CPEs) without EGFP expression were not detected or were limited, suggesting that the occurrence of gene deletion and contamination with the original virus (LC16m8) in the working virus stock was negligible or insignificant.

The initial working stocks of the recombinant LC16m8s, namely LC16m8 carrying NiV G (LC16m8-G) and LC16m8 carrying NiV F (LC16m8-F), were confirmed to express the NiV G and F proteins in the infected cells by IFA using antibodies against G and F, respectively (Fig 2A and 2B). Moreover, the formation of multinucleated giant cells was notably observed in RK13 cells transfected with plasmid encoding the NiV-G or NiV-F genes under the VV early and late promoters (pRecB5R.1-NiV-G or pRecB5R.1-NiV-F) after infection with recombinant LC16m8-F or LC16m8-G, respectively (Fig 2C). In addition, syncytium formation induced by LC16m8 was not significant in RK13 cells. This suggests that the NiV glycoprotein expressed in cultured cells infected with LC16m8-G or LC16m8-F retained its original function of inducing membrane fusion. Although we attempted to establish recombinant LC16m8 expressing both the NiV G and F proteins simultaneously, no virus clones were obtained after the initial homologous recombination step (Fig 1).

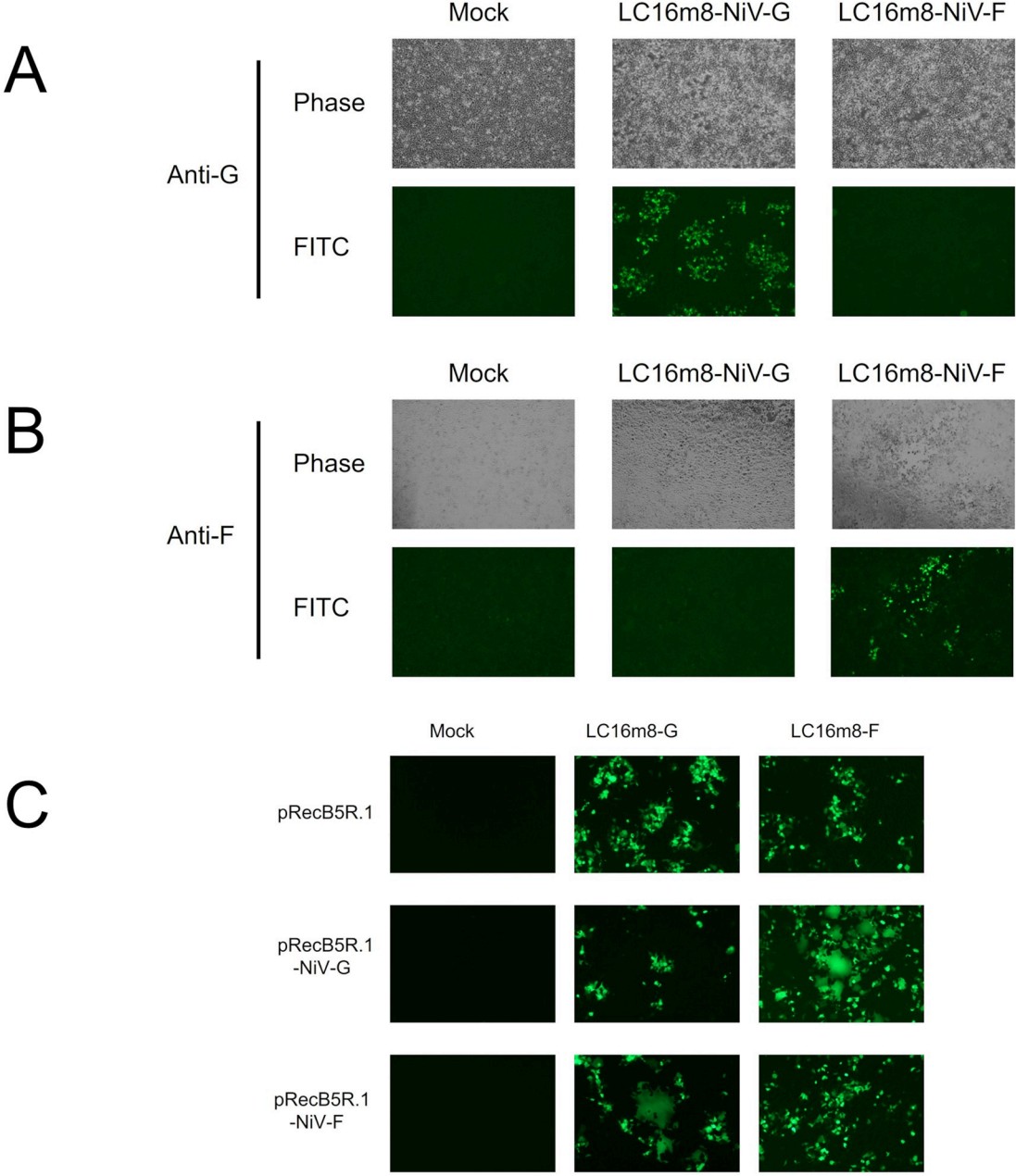

**Fig 2. Expression and function of the envelope glycoproteins of NiV.** (A and B) RK-13 cells were infected with LC16m8-G or LC16m8-F at an MOI of 0.01. At 40 hpi, the cells were fixed and stained with a polyclonal antibody against NiV G protein (A) or F protein (B), followed by incubation with Alexa Fluor 488-conjugated anti-rabbit secondary antibody. The stained cells were observed under a phase-contrast or fluorescence microscope. (C) RK-13 cells were transfected with an expression plasmid driven by the vaccinia early and late promoter (pRecB5R.1-NiV-G, pRecB5R.1-NiV-F, or pRecB5R.1). At 24 h posttransfection, the transfectant cells were infected with LC16m8-G or LC16m8-F at an MOI of 0.01. At 24 hpi, the formation of syncytia was observed under a fluorescence microscope.

## Immunization of hamsters with LC16m8-G or LC16m8-F alone or in combination at one site

Four weeks after the inoculation of hamsters with the VV strain LC16m8 ($1 \times 10^6$ plaque forming unit (PFU)/hamster), three groups of hamsters (VV-G-single-site, VV-G/F-single-site, and VV-mock-single-site) (N = 4, 6, and 7) were intramuscularly inoculated with $5 \times 10^6$ PFU of LC16m8-G, a mixture of LC16m8-G and LC16m8-F ($2.5 \times 10^6$ PFU each), and DMED-10FBS growth medium, respectively (Fig 3A). Two weeks post-inoculation, serum samples were collected from half of the hamsters in each group, and inoculation with the same recombinant viruses or negative controls was repeated as a booster immunization for the remaining

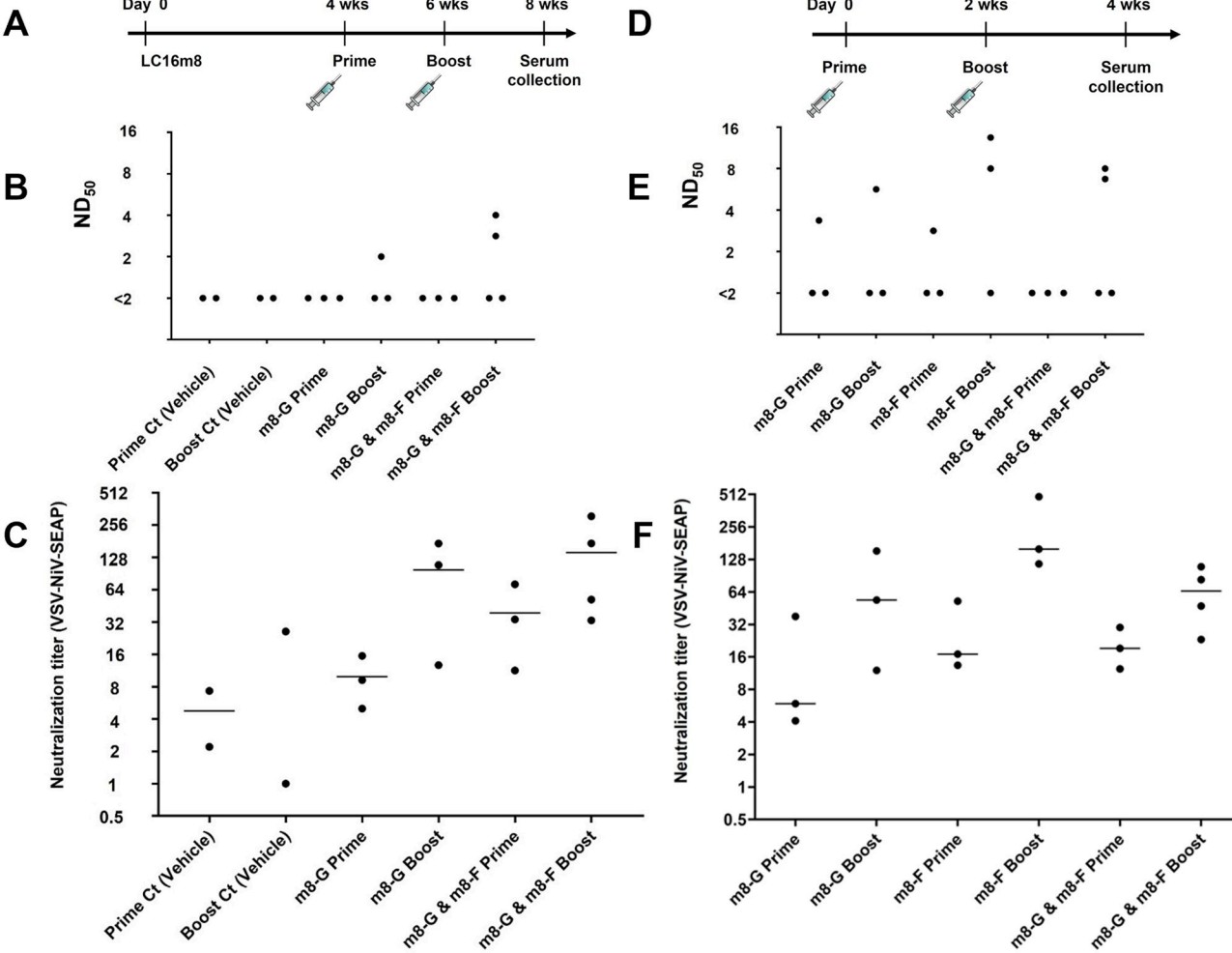

**Fig 3. Induction of neutralizing antibodies against NiV.** (A) Experimental design and schedule of inoculations with LC16m8 and the recombinant LC16m8s. Seventeen hamsters were intramuscularly inoculated with $1 \times 10^6$ PFU of the LC16m8. At 4 weeks, three groups of animals consisting of 4–7 hamsters each were inoculated with $5 \times 10^6$ PFU of the LC16m8-G, $2.5 \times 10^6$ PFU each of LC16m8-G and LC16m8-F, or the growth medium (negative control) as a prime immunization. Half of each group (2–4 hamsters) were further inoculated with the same recombinant viruses as a boost immunization. Two weeks after the final dose of the recombinant LC16m8, all hamsters were euthanized, which was followed by serum collection. Neutralizing titers were analyzed using live NiV (B) or pseudo-typed VSV expressing secreted alkaline phosphatase (SEAP) bearing NiV G and F proteins (VSV-NiV-SEAP) (C). (D) Experimental design and schedule of inoculations with the recombinant LC16m8s. Single recombinant virus (LC16m8-G or LC16m8-F, $5 \times 10^6$ PFU/hamster) or a mixture of the viruses ($2.5 \times 10^5$ PFU each) was intramuscularly administered to hamsters once or twice at a 2-week interval without pre-immunization of LC16m8. Two weeks after the last inoculation with the recombinant virus, sera were collected from all hamsters. Neutralization titers were determined based on the assay using live NiV (E) or the VSV-NiV-SEAP (F). Bars show the mean values.

hamsters in each group. Two weeks after the second inoculation, the serum samples were collected and tested using a neutralizing assay. Using an infectious NiV or vesicular stomatitis virus (VSV) pseudovirus expressing the NiV G and F proteins (VSV-NiV-SEAP) [47], the titer of the neutralizing antibody (SNT or pVSV-SNT) was determined (Fig 3B and 3C).

Two immunizations of the VV-preimmunized hamsters with LC16m8-G barely induced neutralizing antibodies to NiV (Group VV-G-single-site) (Fig 3B), whereas pVSV-SNT was detected by the surrogate assay (Fig 3C). In addition, neutralizing antibodies were not detected in hamsters inoculated with vehicle controls (Group VV-mock-single-site) (Fig 3B), although a non-specific reaction was detected in hamsters inoculated with VSV-NiV-SEAP (<32) (Fig 3C). These results suggest that preexisting immunity against VV greatly impairs the antibody response against NiV. The induction of SNT and pVSV-SNT by the mixture of LC16m8-G and LC16m8-F in the VV-G/F-single-site group was slightly higher than that by LC16m8-G in the VV-G-single-site group, although no statistically significant differences were observed (Fig 3B and 3C).

To evaluate the immunogenicity of LC16m8-F and LC16m8-G more precisely, 19 hamsters were inoculated with LC16m8-G (Group G-single-site, N = 6), LC16m8-F (Group F-single-site, N = 6), or a mixture of LC16m8-G and LC16m8-F (Group G/F-single-site, N = 7) once or twice at 2-week intervals without prior LC16m8 inoculation (Fig 3D). Two weeks after the final inoculation, serum samples were collected, and SNT and pVSV-SNT levels were determined (Fig 3E and 3F). All SNTs detected after the booster immunization shown in Fig 3E were higher (>4) than those detected in Fig 3B. However, SNTs were still below the limit of detection (<2) in one or two serum samples collected after the booster immunization in each group, indicating that neither LC16m8-G nor LC16m8-F efficiently neutralized NiV. The induction of both SNTs and pVSV-SNTs by LC16m8-F appeared to be slightly higher than that by LC16m8-G, although the difference was not significant (Fig 3E and 3F).

None of the hamsters showed any clinical signs, including pox formation or skin lesions, after the administration of LC16m8 or recombinant LC16m8s (Fig 3A and 3D).

## Inoculation of VV-preimmunized hamsters with LC16m8-G or LC16m8-F alone or both viruses at two separate sites

Since two cycles of immunization with LC16m8-G or LC16m8-F only induced low levels of antibody reactions in hamsters, especially with prior immunization with the parental VV (Fig 3B and 3C), we tested three immunization cycles with the recombinant virus in hamsters preimmunized with LC16m8 (Fig 4A). The results in Fig 3B and 3C indicate that the immune response induced by the mixture of LC16m8-G and LC16m8-F neutralized NiV infectivity more efficiently because the antibodies induced by both proteins might function synergistically. However, no synergistic effects were observed or confirmed in subsequent experiments (Fig 3E and 3F). Here, we hypothesized that syncytium formation caused by the NiV-G and NiV-F proteins expressed in cells following infection with LC16m8-G and LC16m8-F would detrimentally affect the induction of antibody reactions. Thus, we tested another inoculation method in hamsters, which included simultaneous inoculation with LC16m8-G and LC16m8-F at different sites (Fig 4A, iii). Four weeks after the preceding immunization of 27 hamsters with LC16m8 ($1 \times 10^6$ PFU/hamster), eight and nine of the 27 hamsters were inoculated with $2 \times 10^6$ PFU of LC16m8-G and LC16m8-F intramuscularly in the right hind limb, respectively (Group 1: VV-G-single-site, and Group 2: VV-F-single-site) (N = 8 and 9) (Fig 4A). The remaining 10 hamsters were inoculated with $1 \times 10^6$ PFU of LC16m8-G and the same titer of LC16m8-F separately but simultaneously in the right and left hind limbs (Group 3: VV-G-F-separate-sites). One-third of the hamsters in each group were further inoculated once

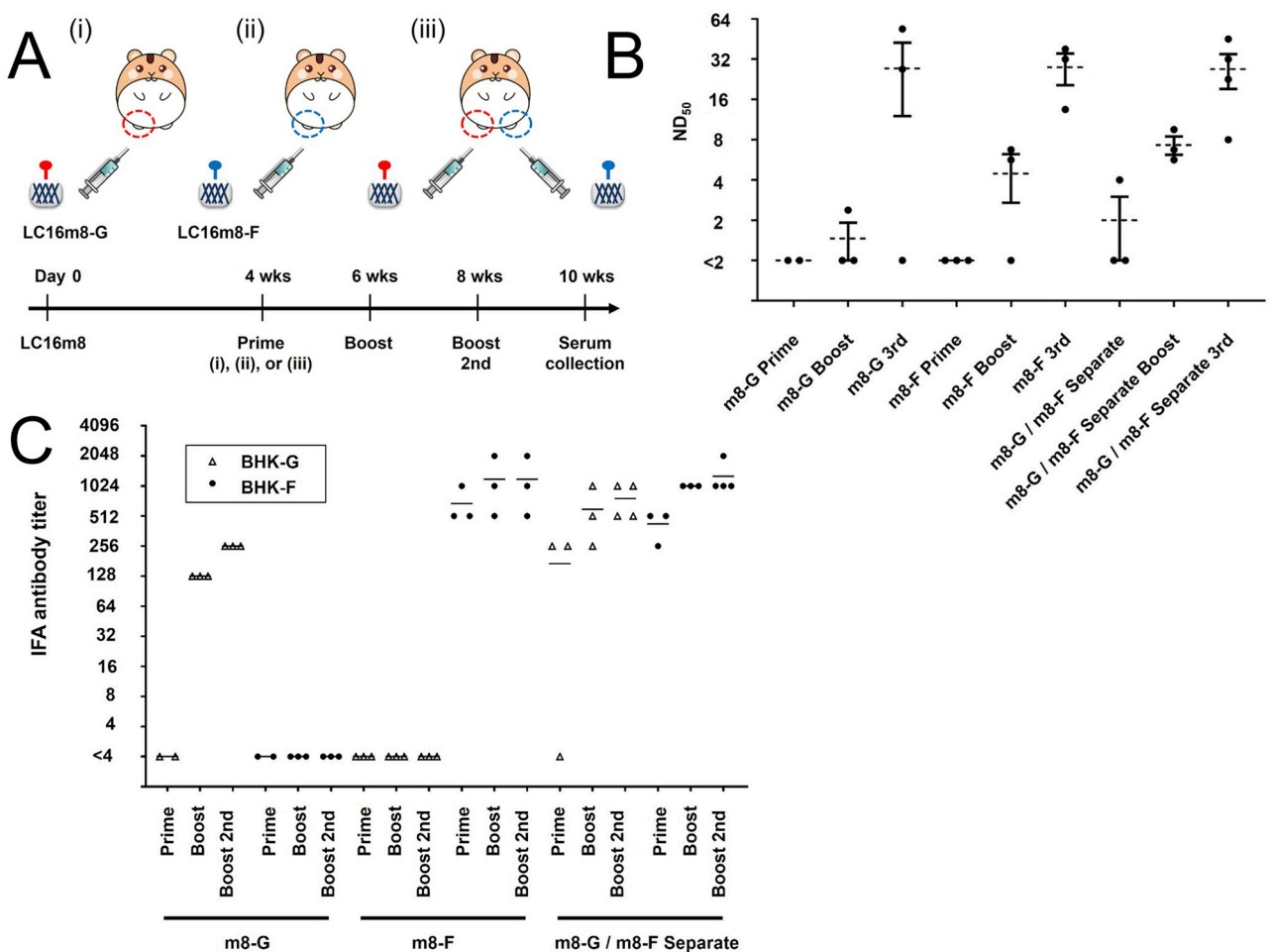

**Fig 4. Effective induction of neutralizing antibodies by three immunization cycles with the recombinant LC16m8s in the VV-preimmunized hamsters.** (A) Experimental design and schedule of inoculations with LC16m8 and the recombinant LC16m8s. Twenty-seven hamsters were intramuscularly inoculated with $1\times10^6$ PFU of the LC16m8. At 4 weeks, three groups of hamsters (n = 8, 9 or 10) were inoculated with $2\times10^6$ PFU of the LC16m8-G (Group 1: VV-G-single-site), $2\times10^6$ PFU of the LC16m8-F (Group 2: VV-F-single-site) or both viruses ($1\times10^6$ PFU each, onto two sites) (Group 3: VV-G-F-separate-sites). Two weeks after infection with the recombinant LC16m8s, one-third of the hamsters in each group were inoculated once or twice at a 2-week interval with the same recombinant viruses as the prime inoculation. Two weeks after the last shot, a serum sample was collected from each hamster. All groups of hamsters are shown with roman numerals in parentheses. Neutralizing titers were determined using live NiV (B). Each circled dot represents the titer ($ND_{50}$) of each serum sample. Dotted line and bars show mean ± SEM. A one-tailed Student's t test assuming unequal variance was used for analyzing the data. $ND_{50}$ values below the LOD ($<2$) were assigned a value of 1. Antibody titers were also determined using the IFA method (C). BHK cells were transfected with the expression plasmid encoding NiV-G or NiV-F. At 24–36 hpi, the cells were fixed and reacted with dilutions of the hamster serum, followed by incubation with Alexa Fluor 488-conjugated anti-hamster secondary antibody. The stained cells were observed under a fluorescence microscope. IFA titers were defined based on the maximum serum dilution in which a fluorescence signal was detected, and the titers measured in BHK cells expressing NiV-G (BHK-G) or BHK cells expressing NiV-F (BHK-F) are indicated by a triangle or circle dot, respectively.

or twice in the same manner as the initial inoculation with the same recombinant LC16m8s. Two weeks after the last inoculation, serum samples were collected at the designated time points and subjected to neutralization tests (Fig 4A). Although three immunization cycles with LC16m8-G induced strong antibody reactions in two hamsters in Group 1 ($>16$), one remaining hamster did not respond to the inoculations (Fig 4B). In contrast, neutralizing antibodies were induced in all three hamsters after three shots with LC16m8-F. However, two injections of LC16m8-F did not induce a reaction in one of the three hamsters in Group 2. In contrast,

double or triple inoculation with the simultaneous inoculation method using LC16m8-G and LC16m8-F at different sites efficiently induced neutralizing antibodies against NiV and produced high titers in all hamsters in Group 3 (Fig 4B). These results indicate that some hamsters immunized with LC16m8-G or LC16m8-F once or twice did not produce any neutralizing antibodies in their sera. Here, we examined whether antibodies (other than neutralizing antibodies) were induced in the sera of these hamsters (Fig 4C). BHK cells transiently expressing NiV G or F proteins were incubated with dilutions of hamster serum, and antibodies against both proteins were detected using IFA. Although the IFA antibody against the G protein was not detected in hamsters that received a single immunization with LC16m8-G, antibodies (128 and 256) were detected in hamsters with two and three cycles of immunization with LC16m8-G, respectively (Fig 4C).

## Discussion

In this study, we succeeded in generating recombinant LC16m8s (LC16m8-G and LC16m8-F) and confirmed the expression of G or F proteins in cells infected with either recombinant LC16m8s (Figs 1 and 2). Moreover, neutralizing antibodies against NiV were induced in all hamsters inoculated three times with these recombinant viruses (Fig 4B). In parallel with the increase in neutralizing antibodies during the three immunizations, an increase in IFA antibody titers was also observed (Fig 4C). However, the levels of neutralizing antibodies induced by one or two inoculations of each recombinant LC16m8 were relatively low or below the detection limit, although IFA antibodies were detected in most hamsters except for the hamsters inoculated once with LC16m8-G (Fig 4C). The levels of neutralizing antibodies induced by LC16m8-based vaccine candidates in the present study were similar to those induced in hamsters inoculated with the NYVAC-based vaccine expressing the NiV glycoprotein in a previous study [13]. However, the number of viruses in the vaccination solution was slightly higher in the previous study (e.g., $2 \times 10^6$ viruses were immunized in the experiment shown in Fig 4B, whereas $10^7$ viruses were immunized in the previous study). Moreover, pre-immunization with VV was performed in the current study, as shown in Fig 4B, but not in the previous study [13]. Although we had already demonstrated that the LC16m8-based vaccine for SFTSV provided protection against its lethal challenge, the protection rate was decreased in the VV-preimmunized mouse [39]. This suggests that pre-existing immunity to VV inhibits the induction of antibodies to foreign antigens (such as NiV glycoproteins) expressed in VV-based recombinant vaccines. Furthermore, neutralizing antibody measurements using infectious NiV were evaluated by complete inhibition of the cytopathic effect in the present study, whereas 50% plaque reduction was used in the previous study (i.e., the value of antibody titers determined by the methods used in the previous study should be higher than those in the current study in principle) [13]. Taken together, these facts suggest that the LC16m8-based vaccine is more effective than the NYVAC-based vaccine at inducing neutralizing antibodies, although strict comparisons are difficult under the conditions of different methods. Two cycles of inoculation with NiV G-expressing NYVAC (NYVAC-NiV-G) or NiV F-expressing NYVAC (NYVAC-NiV-F) have been shown to induce complete protection against lethal NiV infections in hamsters [13]. This suggests that recombinant LC16m8 developed in this study has the potential to protect hamsters from lethal NiV infection after only two cycles of immunization, similar to the NYVAC-based vaccine. When the number of immunization cycles was increased to three, NTs were elevated (more than 16-fold) after the last cycle. As SNTs were determined or measured with complete inhibition of the cytopathic effect in this study, hamsters inoculated with the three immunizations of recombinant LC16m8s were expected to survive after a lethal infection with NiV, even if predicted or anticipated only in terms of the

humoral immune response. However, further challenge studies with NiV are necessary to confirm the efficacy of recombinant LC16m8s as a NiV vaccine.

Originally, LC16m8 grew less efficiently in hamsters and mice than in humans [27]. LC16m8 inoculation induces local skin reactions and erythema in humans [33], whereas it does not induce any clinical symptoms, including skin reactions, in hamsters and mice in the current and previous studies [39]. Rabbits are known to be highly susceptible to VV, but attenuated VV strains, including LC16m8, induce only mild skin lesions, whereas highly virulent strains strongly induce skin lesion formation [20,49,50]. Thus, it has been postulated that there is a correlation between the proliferative potential of VV strains in animals and the degree of skin lesions induced at the inoculation site. All the facts suggest that hamster and mouse models underestimate the potency of recombinant LC16m8 in inducing neutralizing antibodies because of the limited growth features of the virus vector in these animals.

MVA lost approximately 10% of the genome of the parental Ankara strain, especially at both end regions of the genome [51,52]. As a result, MVA does not grow in most cell cultures, except in chicken embryo fibroblasts and BHK-21 cells [29–31,53]. Furthermore, the defective regions contain several genes associated with evasion of the innate immune system, thus leading to the enhancement of antigen presentation and immunogenicity and efficient induction of acquired immunity. These characteristics of MVA may improve vaccine safety and efficacy. The MVA is the most widely used platform for VV-based vaccines in clinical practice. In the NYVAC strain, 18 open reading frames from the parental Copenhagen strain were deleted, and these deleted genes were involved in pathogenicity, virulence, and host range [34]. Compared with the MVA strain, the growth profile of NYVAC in cell culture was comparable or inferior, thus showing similar proliferative features [53]. In contrast, LC16m8 maintains proliferative activity in some cultured cells, including primary rabbit kidney cells, and has a higher proliferative potential than MVA and NYVAC [24]. Compared with the parental strain, LC16m8 lacks the B5R protein (one of the major neutralizing antigens) because of a frameshift mutation with a single gene deletion [28]. Although genetic defects are thought to be one of the main causes of attenuation in the LC16m8 strain, the mechanism of attenuation raises concerns regarding reversion to the parental strain through reversion mutations. Nevertheless, clinical trials conducted in Japan have demonstrated that its safety has been maintained [23,33]. Importantly, LC16m8-G and LC16m8-F deleted the B5R gene region and replaced it with NiV G or F, thereby preventing reversion to the parental strain. Another concern about the decreased induction of neutralizing antibodies is associated with the loss of neutralizing antigens, although LC16m8 has been reported to maintain the ability to induce humoral and cellular immunity [54]. Notably, in studies reporting the efficacy of LC16m8 or recombinant LC16m8, the input of viral doses is usually 10–100 times lower than that of MVA or NYVAC [39,50,54]. Taken together, LC16m8 is a promising viral vector vaccine format that can induce a stronger humoral immune response based on its relatively high or modest proliferative potential compared to other attenuated VV-based vaccines. In addition, an attenuated canary poxvirus expressing the glycoprotein of Hendra virus (a closely related species to NiV) has been reported [14]. Canary poxviruses have low proliferative potential in host animals, except for avian species. All the facts suggest that the LC16m8-based vaccine format has superior features as a proliferative vaccine compared with other poxvirus-based vaccines.

In the present study, the antibody titers induced by LC16m8-G tended to be lower than those induced by LC16m8-F, although no significant differences were detected. This trend is consistent with previous reports on NYVAC-based vaccines [13]. Interestingly, although LC16m8-G may be less effective than LC16m8-F in inducing neutralizing antibodies, the most robust induction of neutralizing antibodies was observed in hamsters inoculated simultaneously with LC16m8-G and LC16m8-F, but not in hamsters inoculated with LC16m8-F

alone (Fig 4B). Notably, LC16m8-G, F, which carry both G and F genes, were not rescued in this study (Fig 1). These facts suggest that the growth of LC16m8 is inhibited in to the local site where G and F are expressed, and syncytia formation is induced. However, further studies are required to verify this possibility.

## Acknowledgments

We thank Ms. Momoko Ogata of the National Institute of Infectious Diseases of Japan for her assistance in this study. We are grateful to Dr. Kouichi Morita of Nagasaki University for providing the NiV. We thank Drs. Jun Arii and Yuta Shirogane for their valuable comments. We would like to thank Editage (www.editage.com) for the English language editing.

## Author Contributions

**Conceptualization:** Shumpei Watanabe, Tomoki Yoshikawa, Yoshihiro Kaku, Takeshi Kurosu, Hikaru Fuji, Shigeru Morikawa, Masayuki Shimojima, Masayuki Saijo.

**Formal analysis:** Shumpei Watanabe, Yoshihiro Kaku.

**Funding acquisition:** Shumpei Watanabe, Masayuki Shimojima.

**Investigation:** Shumpei Watanabe, Tomoki Yoshikawa, Yoshihiro Kaku, Takeshi Kurosu, Shuetsu Fukushi, Satoko Sugimoto, Yuki Nishisaka, Masayuki Shimojima.

**Project administration:** Shumpei Watanabe, Ken Maeda, Hideki Ebihara, Masayuki Shimojima, Masayuki Saijo.

**Resources:** Tomoki Yoshikawa, Glenn Marsh.

**Supervision:** Masayuki Shimojima.

**Visualization:** Shumpei Watanabe, Yoshihiro Kaku.

**Writing – original draft:** Shumpei Watanabe, Masayuki Saijo.

**Writing – review & editing:** Shumpei Watanabe, Tomoki Yoshikawa, Yoshihiro Kaku, Takeshi Kurosu, Shuetsu Fukushi, Satoko Sugimoto, Yuki Nishisaka, Hikaru Fuji, Glenn Marsh, Ken Maeda, Hideki Ebihara, Shigeru Morikawa, Masayuki Shimojima, Masayuki Saijo.

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
