## [Decision Letter · Decision Letter 0]

25 Jul 2023

Dear Dr Watanabe,

Thank you very much for submitting your manuscript "Construction of a recombinant vaccine expressing Nipah virus glycoprotein using the replicative and highly attenuated vaccinia virus strain LC16m8" for consideration at PLOS Neglected Tropical Diseases. As with all papers reviewed by the journal, your manuscript was reviewed by members of the editorial board and by several independent reviewers. In light of the reviews (below this email), we would like to invite the resubmission of a significantly-revised version that takes into account the reviewers' comments. 

We cannot make any decision about publication until we have seen the revised manuscript and your response to the reviewers' comments. Your revised manuscript is also likely to be sent to reviewers for further evaluation.

Sincerely,

Prashant Kumar, Ph.D.

Academic Editor

Andrea Marzi

Section Editor

Reviewer's Responses to Questions

**Key Review Criteria Required for Acceptance?**

**Methods**

-Are the objectives of the study clearly articulated with a clear testable hypothesis stated?

-Is the study design appropriate to address the stated objectives?

-Is the population clearly described and appropriate for the hypothesis being tested?

-Is the sample size sufficient to ensure adequate power to address the hypothesis being tested?

-Were correct statistical analysis used to support conclusions?

-Are there concerns about ethical or regulatory requirements being met?

Reviewer #1: Specific comments for the methods are detailed below

Reviewer #2: Objectives are clearly stated with a clear testable hypothesis stated. The study design appropriately addresses the stated objectives. 

Hamsters are well-documented animal models for human Henipavirus infection and are appropriate for the hypothesis being tested. Three (3) groups of hamsters (Group G-single-site, F-single-site, and G/F-single-site) were inoculated to evaluate antibody induction by the recombinant viruses (LC16m8-G, LC16m8-F). Similarly, three (3) groups of 

hamsters (Group VV-G-single-site, VV-G/F214 single-site, and VV-mock-single-site) harboring pre-existing immunity against VV were inoculated. 

However, the authors did not indicate the number of Hamsters used for each group. Authors could state, for example, that 2 hamsters each were inoculated for each group. This should be stated in the Animal Experiment section, and not just the figure legends.

Statement in Lines 206-210 under the subheading 'Animal Experiment' is repeated in Lines 210-214.

Appropriate statistical analysis was used, and there were no ethical concerns.

**Results**

-Does the analysis presented match the analysis plan?

-Are the results clearly and completely presented?

-Are the figures (Tables, Images) of sufficient quality for clarity?

Reviewer #1: Specific comments for the results section are detailed below

Reviewer #2: Analysis presented match the analysis plan, and results are clearly and completely presented.

**Conclusions**

-Are the conclusions supported by the data presented?

-Are the limitations of analysis clearly described?

-Do the authors discuss how these data can be helpful to advance our understanding of the topic under study?

-Is public health relevance addressed?

Reviewer #1: Specific comments for the discussion section are detailed below

Reviewer #2: Conclusions are supported by the data presented with clearly stated limitations. The successful induction of neutralising antibodies against Nipah virus (NiV) in hamsters using LC16m8-based vaccine format provides an important basis for the clinical use of all vaccinia virus-based vaccines against NiV disease. This is well articulated by the authors

**Editorial and Data Presentation Modifications?**

Reviewer #1: the comments below pertain to the different sections of the manuscript:

Specific comments

L9-14: Complete Affiliation of these footnotes should be mentioned - city, prefecture, etc.

L32: I suggest that full name will be included when mentioning the viral proteins for the first time in the text.

L60-61: Any reference?

L89: I suggest adding some reference to the recent monkeypox outbreak that brought to attention the use of new VV-based vaccines, while addressing the undesired side effects.

L123: I suggest adding "G or F GLYCOPROTEINS" or ANTIGENS for clarity

L198: The number of animals in each group should be mentioned clearly to allow the reader evaluate the statistical significance of the results.

L207: what does this mean? they were vaccinated with VV earlier? can the authors add some reference or refer the reader to a description of the background of the animals in this context?

L274: I think that rephrasing to "expression of mCherry under conditional drug selection", or something like that, and avoid using the word "influence" in that context.

 L276: I suggest that the authors will give the full name of the reagent, at least in the first time it 

is mentioned in the text.

L306: How many animals were in each group?

 L311: this was a booster? if so, please mention it. if not, please explain the purpose of this step.

L315L I think that either "bearing...the...genes" or "expressing...the....proteins" should be used here.

L332: the tests were PERFORMED not DETERMINED. the neutralization titer was determined.

L349: I think it should be Fig 4, not 3

L389: Figure 4C shows the average and SEM of each group, not individual animals. how can the reader understand that one animal in the G group did not produce anti-NiV G antibodies?

L408: if the authors performed physical inspection to identify any such reactions in the tested animals in this study, they should report this in the results. otherwise, it should be stated that the animals were inspected regularly and did not show any clinical signs of adverse reaction to the inoculation. 

L410: was there any experiment that measured the viral load in the animals? either titration or quantitative PCR? if so, please refere the reader to the appropriate data. if not, this statement needs a different reference.

L428: Does this refer to human clinical trials? is the LC16m8 used, or was it tested on humans? the authors should be very clear on the safety profile of this strain if the argue that it can be used as an alternative to other VV scaffolds for recombinant vaccines.

L433: a very low RATE - or - very low RATES

L443: i did not understand whether the LC16m8 replicates in humans or not. If it does, than adverse side effects, secretion and reversion to virulence are a noticable concern. Recently, an attenuated VV strain was used for mokeyopx vaccination, which does not replicate in humans, and it is therefore much safer for immunocompromised patients ( DOI: 10.1056/ nejmoa2215201). I think that the authors should address the option of using a single-cycle strain, such as the JINNEOS vaccine, for a safer vaccine scaffold.

Figure Legends

L711 (Fig.1): is there a particular reason that the selection and indicator genes were designed in an opposite orientation to the GOI? i am curious to know, as this might make the cloning less conveinet, but there is no specific explanation for that.

L767: average and SEM of how many repeats? were technical repeats for each serum sample? or one well/plate for each sample? please provide details on the experiment and number of repeats.

Reviewer #2: Accept with minor revision Authors should indicate the number of hamsters used for each group under the 'Animal Experiment' subheading.

**Summary and General Comments**

Reviewer #1: The study described in this manuscript is definitely important and address an important need to help the efforts to overcome and prevent Nipah virus infections. There are some issues, however, that I think need to be addressed by the authors to render the manuscript ready for publication in PNTD.

The authors claim that the constructs they produced are superior to those developed by Guillaume et al. in 2003 (doi: 535 10.1128/jvi.78.2.834-840.2004.). while in the 2003 study, the authors demonstrate actual protection in a challenge experiments, both after immunization and as passive immune response (antibody administration), in the present manuscript, no such challenge experiment is described. The authors indeed show that immunization with both G- and F-expressing recombinant viruses induces significant neutralizing antibodies, but in the absence of a challenge experiment, their argument, in my opinion, is not firm. In order to support the major claim of this study, i.e., convince that the LC16m8 NiV vaccine is a promising alternative, I suggest that the authors will include a more detailed comparison of their results to those of Guillaume et al., for the parts that were performed in both studies. This way, the reader can appreciate the outcomes of the two studies in a comparative manner. 

Also, I suggest that the authors will revise the relevant parts of the discussion section, to incorporate a part that addresses the limitations of the present study, including the number of animals, the absence of a comprehensive safety study, and the absence of a challenge experiment. 

It is somewhat difficult to argue that the new vaccine scaffold is superior without a challenge experiment to demonstrate it, so a more detailed comparison of the existing data might strengthen it nonetheless.

Another issue is the complete dataset. The authors show in some cases average values, and I think that a supplementary table or tables with the values obtained for each animal, depending on the experiment, will be helpful for the reader to appreciate the work. Additionally, this is a requirement of the journal.

As I mention below in the specific comments regarding the discussion section, I suggest that the authors will address the option of using a non-replicative VV scaffold, such as the one used for the recent JYNNEOS vaccine, as a safer option. Since it cannot replicate in the human body, it has significantly fewer and less severe side effects and contraindications, compared with previous VV-based vaccines, some of which are mentioned in this manuscript.

Lastly, although the language is good overall, I noticed some sentences that need to be rephrased or corrected. It is likely that I missed some others, so a second language check might be useful before submitting a revised version.

In conclusion, I think that the manuscript could be suitable for publication in PNTD, after the authors will address the points raised in this review.

Reviewer #2: The study looks good and original with public health relevance. Laboratory experiments were carefully carried out and in detail. The manuscript was well written.

PLOS authors have the option to publish the peer review history of their article (what does this mean?). If published, this will include your full peer review and any attached files.

Reviewer #1: No

Reviewer #2: Yes: Lawrence Annison
---

## [Decision Letter · Decision Letter 1]

7 Dec 2023

Dear Dr Watanabe,

We are pleased to inform you that your manuscript 'Construction of a recombinant vaccine expressing Nipah virus glycoprotein using the replicative and highly attenuated vaccinia virus strain LC16m8' has been provisionally accepted for publication in PLOS Neglected Tropical Diseases.

Best regards,

Prashant Kumar, Ph.D.

Academic Editor

Andrea Marzi

Section Editor

---

## [Editor Report · Acceptance letter]

12 Dec 2023

Dear Dr Watanabe,

We are delighted to inform you that your manuscript, "Construction of a recombinant vaccine expressing Nipah virus glycoprotein using the replicative and highly attenuated vaccinia virus strain LC16m8," has been formally accepted for publication in PLOS Neglected Tropical Diseases.

Best regards,

Shaden Kamhawi

co-Editor-in-Chief

Paul Brindley

co-Editor-in-Chief
